# Improving Barrier Properties of Xylan-Coated Food Packaging Papers with Alkyl Ketene Dimer

**Petronela Nechita** [1],*, **Mirela Roman** [2],*, **Alina Cantaragiu Ceoromila** [3] and **Andreea Veronica (Dediu) Botezatu** [4]

1. Department of Environmental, Applied Engineering and Agriculture, Dunarea de Jos University of Galati, 817112 Braila, Romania
2. Doctoral School of Fundamental and Engineering Sciences, Dunarea de Jos University of Galati, 817112 Braila, Romania
3. Cross-Border Faculty, Dunarea de Jos University of Galati, 47 Domneasca Street, 800008 Galati, Romania
4. Department of Chemistry Physical and Environment, Faculty of Sciences and Environment, Dunarea de Jos University of Galati, 800201 Galati, Romania
* Correspondence: petronela.nechita@ugal.ro (P.N.); mirela.roman@ugal.ro (M.R.)

**Abstract:** In order to improve the hydrophobicity of xylan hemicellulose, a simple procedure of its chemical modification with alkyl ketene dimer (AKD), a non-toxic, cost-effective, and eco-friendly chemical, was performed. For this purpose, the reaction products of beech wood xylan and different amounts of hydrophobic AKD were used for paper surface treatment. Thus, the coatings of about 4.5 g/m$^2$ were applied on both sides of base paper in single and three successive layers. To obtain a complete reaction between AKD and xylan hemicellulose, the coated papers were thermal cured (about 110 °C) and the effects of AKD content on the barrier (water, oil, and water vapours) and mechanical properties were analysed. The structural analyses by scanning electron microscopy (SEM) and Fourier transform infrared spectroscopy (FT-IR) of coated samples emphasized the presence of β-keto-ester compounds as a result of the reaction between xylan hemicelluloses and AKD. This is confirmed by the improving of barrier properties as the AKD content in coating dispersion is higher. The good barrier performance and improved strength properties were obtained for the coated papers with xylan hemicellulose and 1% AKD applied on paper surface in three successive layers (about 4.5 g/m$^2$). In this case, the water vapours transmission rate (WVTR) was 35% lower than those untreated and the resistance to air passing through coated papers was over 3 times higher compared with base paper. There are no results reported on the chemical reaction of xylan hemicelluloses with AKD as well as its application in coatings for paper packaging. In this context, the obtained results in this study can contribute to expand the applications area of hemicelluloses offering a sustainable strategy for the developing of food packaging papers with appropriate barrier properties using biopolymer coating materials.

**Keywords:** hemicellulose; xylan; alkyl ketene dimer; paper coatings; food packaging; barrier properties



## 1. Introduction

Currently, when the plastic packaging production and consumption are limited, the interest to develop the high recycling and biodegradable packaging materials is intensified. In this context, the cellulosic materials (paper and board) are considered promising candidates having in view their inherent advantages of being cheaper than other materials, highly recyclable and biodegradable, and easy to convert into packaging boxes with determined strength and stiffness. In addition, paper and board packages are lighter and easy to print, and are more resilient over a wider temperature range in comparison with glass and plastic materials [1].

Besides plastic, the cellulose-based products represent the packaging materials most used for liquids, and dry or fatty foods. The paper and board materials have the highest recycling rates worldwide, accounting for over 70% [2,3].

Generally, most food packaging papers are subject to requirements for good barrier properties and high mechanical strength. The strength properties are mainly controlled by additives based on natural polymers such as starch or cellulose derivatives. Barrier properties, such as water vapour, oxygen or flavours, and microbial attack resistance are very important during the control of different compounds permeation through the packaging material [4]. These characteristics are considered the main parameters in the control of odour and aroma as well as in the protection and extending of food shelf life [5].

However, as food packaging, paper/board has inherent poor barrier properties due to the porous structure and cellulose fibres. Currently, the existing food packaging papers with barrier properties are based either on coatings with petroleum base polymers or they are obtained by lamination process with aluminium and plastic foils [6–9].

Due to increasing environmental protection awareness, the demand for recyclable and biodegradable packaging materials obtained from renewable resources is of high interest and the concept of sustainability is a determining factor for the packaging market. In this context, the focus of research is addressed on identification of alternative solutions to synthetic polymers by using biopolymers included in coatings for food packaging paper [10,11].

Based on their biodegradability and low toxicity, polysaccharides are considered a promising sustainable alternative to replace synthetic polymers in coatings for food packaging paper. Furthermore, the polysaccharides have appropriate affinity for paper substrate being used as additives for the improvement of strength in papermaking. Due to their ability to forma film, the polysaccharides exhibit barrier properties against gases, liquids, and aroma.

After cellulose, the second group of polysaccharides that exist in cell wall plants are hemicelluloses. Generally, hemicelluloses are secondary products of the dissolving pulp process and used as heat energy and biofuel sources or they are converted in chemicals. Based on their properties, the area of application of hemicelluloses can be extended as an appropriate resource of biopolymers in coatings for paper applications [12].

Xylan is the main component of the hemicelluloses class, and is available in hardwood and about 30% in wheat straw. The commercial applications of xylan hemicellulose are limited to the obtaining of xylitol and biofuels by biological conversion of sugar. In the packaging industry, xylan is used as an additive for improving the strength and biodegradability of plastic materials [13–17].

In our previous studies [12], the performance of hardwood xylan hemicellulose in its native form and as polyelectrolytes complexes with chitosan biopolymer has been evaluated in the coating of paper and to obtain films for food packaging. The results emphasized slightly improved barrier properties and moderate antifungal activity on xylan-coated papers. The flexibility and swelling capacity of xylan films were improved by adding chitosan biopolymer. These effects are the result of hydrophilic nature of xylan with a large number of hydroxyl groups in the monomeric unit. However, free hydroxyl groups are available for chemical functionalization of xylan (i.e., esterification, etherification, crosslinking, etc.) with hydrophobic functional groups, especially when it is used for packaging and coating applications [18].

Hydrophobically modified polysaccharides are of high interest from an environmental and practical point of view because they are biobased, and many of them have been reported for packaging applications [19–21]. The esterification is the most frequently used method for chemical modification, and many studies on acetylated xylan have been published [22–24]. The reaction of xylan with propylene oxide and then acetylated, butylated, or allylated hydroxypropyl xylan are other reported approaches [25–27].

In this context, the reaction of xylan with a ketene dimer or anhydrides can be of interest from both fundamental and practical points of view, and continues the research in the field of new types of hydrophobic xylans. In recent decades, some research studies have reported the reaction of alkyl ketene dimer (AKD) with microcrystalline and microfibrillated cellulose [28–30], cellulose nanofibres [31–33], carboxymethyl and hydroxyethyl

cellulose [34,35], starch [36,37], and cashew gum [38]. In other studies [39,40], mixtures of polyvinyl alcohol and alkyl ketene dimer have been used as coatings for packaging paper to improve the barrier properties and paper hydrophobicity.

However, there are no results reported on the chemical reaction of xylan hemicelluloses with AKD as well as its application in coatings for paper packaging.

AKD is the most widely used internal sizing agent in the world for imparting water resistance to paper and paperboard products. In addition, AKD is biodegradable, with low toxicity and is cheaper compared with other hydrophobic additives, and it is highly efficient for the hydrophobization of cellulose fibres as a water-based solution, even in a small quantity [41]. In the paper industry, AKD is suitable for the permanent hydrophobization of many different kinds of paper such as newspaper, printing and writing paper grades, as well as for cardboard used as a container for alimentary liquids packaging (such as milk) [42].

Alkyl ketene dimers are organic compounds which react with the cellulose chains during paper sizing to form covalent bonds [43,44]. AKD consists of two long alkyl chains on a four-membered heterocycle and the main mechanism of the AKD reaction is based on blocking the -OH groups present on the cellulose fibres surface and as result, any hydrogen bond with water is avoided [28,31].

Among these technical and environmental advantages, the commercial availability of AKD strengthens its use in the process of xylan functionalization by chemical modification and thus, a new type of hydrophobic xylan can be commercially produced. This new strategy contributes to extend the area of xylan application to obtaining sustainable packaging products.

The chemical reaction of AKD with xylan hemicelluloses occurs at free hydroxyl groups from 2 and/or 3 positions of xylan monomer unit. As result, the hydrophobic moieties of β-ketoester and fatty acid chains are introduced, and the amphiphilicity as well as the surface activity of hemicellulose are enhanced (Figure 1) [18,45].

**Figure 1.** The chemical reaction of xylan hemicellulose with AKD [18].

Even if it is known that in the papermaking process the AKD size has efficiency at high temperatures, it has been proved that AKD is relatively unreactive towards both water and hydroxyl-containing compounds, including cellulosic fibres, starch products, or other polysaccharides [46]. Therefore, there is a continuing need to obtain the formation of covalent bonds between xylan hemicellulose and AKD that plays an important role with respect to the development of hydrophobicity.

In this context and having in mind the sustainability issues in the field of packaging products, the goal of this research was to identify a new appropriate strategy to reduce the hydrophilic features of xylan hemicellulose through AKD reaction.

Generally, in conventional applications of AKD in papermaking, the efficiency with which the sizing additive is retained on the cellulose fibres depends on the application of very high molecular mass polyelectrolytes known as retention aids and it is also influenced by the presence of mineral particles of filler or other chemical agents used as functional or process additives.

For a better elucidating of some key effects, in this work AKD was used in the relatively pure form to hydrophobize xylan hemicellulose, in the absence of any other influences, and the performance of the new AKD-modified xylan was evaluated in the coating of packaging paper. Therefore, the hardwood xylan hemicellulose was chemically modified through reactions with different amounts of hydrophobic AKD and used for paper surface

treatment. The coated paper samples were thermal cured (about 110 °C) and the effects of AKD content on the barrier (water, oil, and water vapours) and mechanical properties were analysed.

To identify the formation of covalent bonds between hydroxyl groups of xylan hemicellulose and AKD or the presence of ester functional groups, the structural and morphological properties of coated papers samples were performed using Fourier transform infrared spectroscopy and scanning electron microscopy techniques. The uncoated paper (base paper) and a paper sample coated with native xylan only were used as references.

## 2. Materials and Methods

### 2.1. Materials

Commercial paper from unbleached cellulose pulp of 50 g/m$^2$ weight was used as base paper for coating.

Xylan hemicellulose, from beech wood, was purchased from Carl Roth Germany as a powder coloured light beige to brown with a molecular mass of (132)N and loss on drying $\leq$ 10.0%.

Alkyl ketene dimer (AKD), as commercial product Aquapel TM 210D, was purchased from Solenis, USA (milky white liquid, odourless, total solids of 16.2% and viscosity of 4 cPs/t = 25 °C).

### 2.2. Methods

#### 2.2.1. Surface Treatments (Coating) of Paper

The xylan was dried for 24 h in a laboratory oven at 70–80 °C before use. The coating dispersions based on xylan hemicellulose (as 25 g/L water dispersion) and different amounts of AKD (0.2%, 0.5%, 1%, and 1.5%) were obtained by magnetic stirring at 1500 rpm for 3 h. After that the coating dispersion was applied on a paper surface in a single layer on both sides of the paper using the laboratory rod Meyer system (about 4–5 g/m$^2$) as described in a previous work [12]. For each series of coated papers, a number of 20 samples of 20 × 25 cm were obtained and tested regarding the functional properties. Uncoated paper (base paper) and coated paper with xylan only were used for comparison. Based on the level of barrier properties obtained after the paper treatment with the above-mentioned coating formulas, the coatings based on xylan hemicellulose and 1% AKD were applied on the paper surface in three successive layers until obtained 4–5 g/m$^2$ on both sides of the paper. After surface treatment, all samples were dried for about 10 min at room temperature, then in an oven at 60 °C. After oven drying, the coated paper samples were cured at 110 °C by hot pressing to complete the reaction of AKD with xylan hemicellulose. Before characterization, all samples were conditioned for 48 h at 23 °C and 50% RH.

#### 2.2.2. Testing Methods for Coated Paper Samples

- FT-IR analysis

The Nicolet iS50 FT-IR spectrometer (Thermo Scientific, Waltham, MA, USA) equipped with an attenuated total reflection (ATR) accessory and a diamond crystal plate, in transmission mode was used to highlight the presence of specific chemical groups in the tested samples. The spectrometer was placed in a temperature-controlled laboratory (21 ± 2 °C). Infrared spectra were measured in the spectral range 4000–400 cm$^{-1}$ at 2 cm$^{-1}$ spectral resolution and 32 background/sample scans using OMNIC software (Thermo Fisher Scientific Inc., Waltham, MA, USA). The background spectrum was collected by taking air as a reference before each measurement, and the diamond crystal plate was cleaned with alcohol [47].

- SEM analysis

The microstructure of the paper surfaces was investigated by the scanning electron microscopy technique with Quanta 200 system (FEI). SEM images (100 μm and 10 μm scale bars) were captured at 20 kV as an accelerating voltage using secondary electron signal

in low-vacuum conditions. All paper samples were fixed on carbon doubled-sided tape by attachment onto metal support stubs. Then they were coated with a thin metallic layer for better conductivity, using sputtering equipment (SPI Supplies). The morpho-structural characteristics of the tested materials were evaluated at $1.000\times$ and $10.000\times$ magnifications. A representative number of pictures were taken.

- Barrier properties

Water absorption capacity (Cobb60 index), $g/m^2$, was measured according to the standard method described in SR ISO 535: 2014 (Paper and board, determination of water absorptiveness, Cobb method) [48]. Five measurements for each sample were performed. The average value and standard deviation were calculated and used for results interpretation.

Oil absorption capacity (Unger-Cobb600 index), $g/m^2$, was measured according to the standard method T-441 om −98. The paper sample is came into contact with a given amount of rapeseed oil for 600 s and weight differences were compared.

Water vapours transmission rate (WVTR), $g/m^2$.day, was determined as described in the standard method ISO 2528:2018 (Sheet materials, determination of water vapours transmission rate (WVTR), Gravimetric (dish) method) [49]. Dishes containing a desiccant and closed by the paper samples to be tested were placed in a controlled atmosphere (23 °C and 50% RH) for 4 days. At each 24 h, the dishes were weighed and the WVTR was determined as weight increasing. The measurements were performed in triplicate for each sample.

Static contact angle, °, was determined according to standard T-458 cm-04, measured by the static sessile drop method on OCA15EC instrument. For data acquisition and processing, a digital camera and SCA20 software were used. The water drops were deposited with a micro syringe on the surface of paper samples, which were fixed with clamps on the test table. The contact angle value was calculated after 5 s of water–substrate contact. A total of 10 measurements were performed for each sample of coated paper.

Air permeability, s, was evaluated through the Gurley method according to standard ISO 5636-5:2013 (Paper and board, determination of air permeance (medium range), Part 5: Gurley method) [50]. A determined air volume was passed through the paper sample with a given area and time, s, for air passing was recorded. A total of 10 measurements were performed for each sample of coated paper.

- Mechanical properties

Dry and wet tensile indexes, $N\cdot m/g$, were determined as described in the standard method SR EN ISO 1924-2:2009 (Paper and board, determination of tensile properties, Part 2: constant rate of elongation method) and ISO 3781:2011 (Paper and board, determination of tensile strength after immersion in water) [51,52]. The tensile indexes were measured as the maximum tensile strength of the paper sample until breaking point divided by the grammage of the sample. A total of 10 measurements were performed for each sample of coated paper.

Bursting strength index, $kPa\cdot m^2/g$, was determined according with standard method SR EN ISO 2758:2015 (paper, determination of bursting strength) [53]. The maximum hydrostatic pressure required to break the sample paper was divided by the grammage of the sample. A total of 10 measurements were performed for each sample of coated paper.

For all measurements, the average value and standard deviation were calculated and used for results interpretation.

## 3. Results and Discussions

### 3.1. The Structural Properties of Coated Paper Samples by SEM

The SEM images of coated papers with xylan hemicellulose and different contents of AKD are presented in Figure 2.

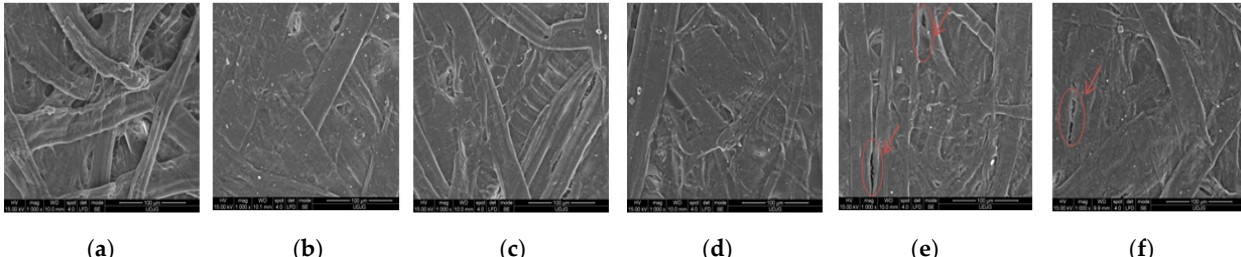

**Figure 2.** SEM images of paper samples coated in a single layer: (**a**) base paper; (**b**) xylan-coated paper; (**c**) xylan + 0.2% AKD-coated paper; (**d**) xylan + 0.5% AKD-coated paper; (**e**) xylan + 1.0% AKD-coated paper; (**f**) xylan + 1.5% AKD-coated paper.

Compared with untreated paper, the paper samples coated with xylan hemicellulose and different contents of AKD exhibit a smooth, homogeneous, and flat surface (Figure 2b–d). For the samples with 1% and 1.5% AKD content, some defects and nonuniformities are observed in the coating layer structure (Figure 2e,f).

In the case of paper samples coated with three successive layers, the defects are more pronounced for xylan-coated papers (Figure 3a). This can be explained by the hydrophilic character of xylan dispersion which easily penetrates the paper substrate compared with more hydrophobic xylan/1% AKD dispersion that leads to obtain a more uniform coated surface with lower defects (Figure 3b). This is the result of hydrophobic nature of AKD that fills the pores of base paper structure even from the first layer (Figures 2e and 3b).

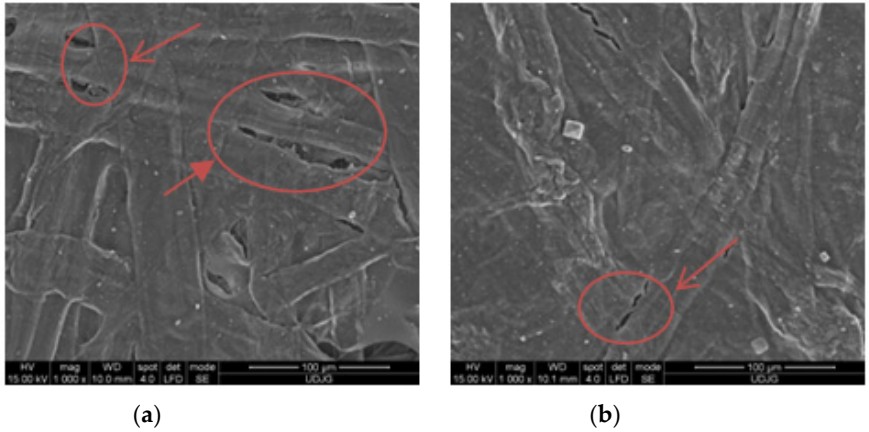

**Figure 3.** SEM images of paper samples coated with three successive layers: (**a**) xylan-coated paper; (**b**) xylan + 1.0% AKD-coated paper.

The improved hydrophobicity of coating dispersions based on xylan hemicellulose and AKD is the result of ketoesters formation after the esterification reaction, with higher intensity as the AKD content is increased (Figure 4a–e).

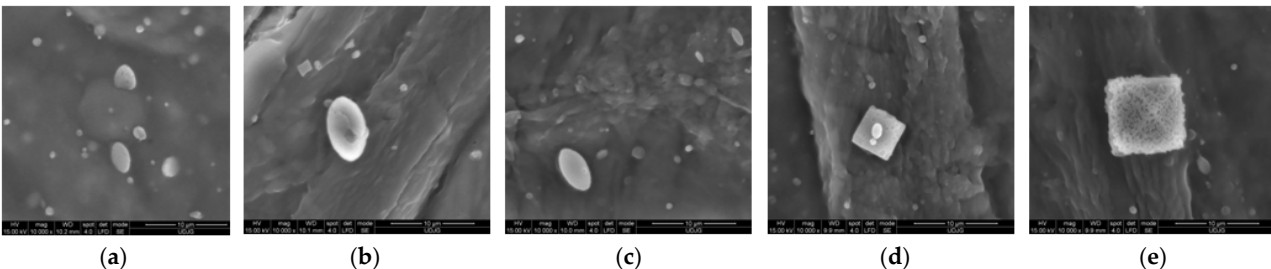

**Figure 4.** SEM images of β-ketoesters formation: (**a**) xylan particles; (**b**) xylan/0.2% AKD; (**c**) xylan/0.5% AKD; (**d**) xylan/1.0% AKD; (**e**) xylan/1.5% AKD.

### 3.2. The Structural Analysis by FT-IR

Xylan hemicellulose was modified via esterification using alkyl ketene dimer and the obtained dispersions were applied on paper surface in order to improve the barrier properties.

The obtained samples were characterized using FT-IR and the typical IR spectra are presented in Figure 5a. The most important modifications appear in the ranges from 2800 to 3500 cm$^{-1}$ and 820 to 1790 cm$^{-1}$. The absorption peak in the range of 3300–3500 cm$^{-1}$ was attributed to the stretching vibrations of the hydroxyl groups from xylan and hydrogen bonds between cellulose fibres [41,54–56]. Within the interval between 820 and 1790 cm$^{-1}$ is presented the stretching absorption of the β-glucosidic bonds between sugar units (896 cm$^{-1}$). The band between 1000 and 1200 cm$^{-1}$ is caused by C=O stretching. Within the range 1602 cm$^{-1}$–1733 cm$^{-1}$ appeared the stretching vibration characteristic to the absorption peaks of β-ketone ester bond formed between xylan hemicellulose and AKD (Figure 5a).

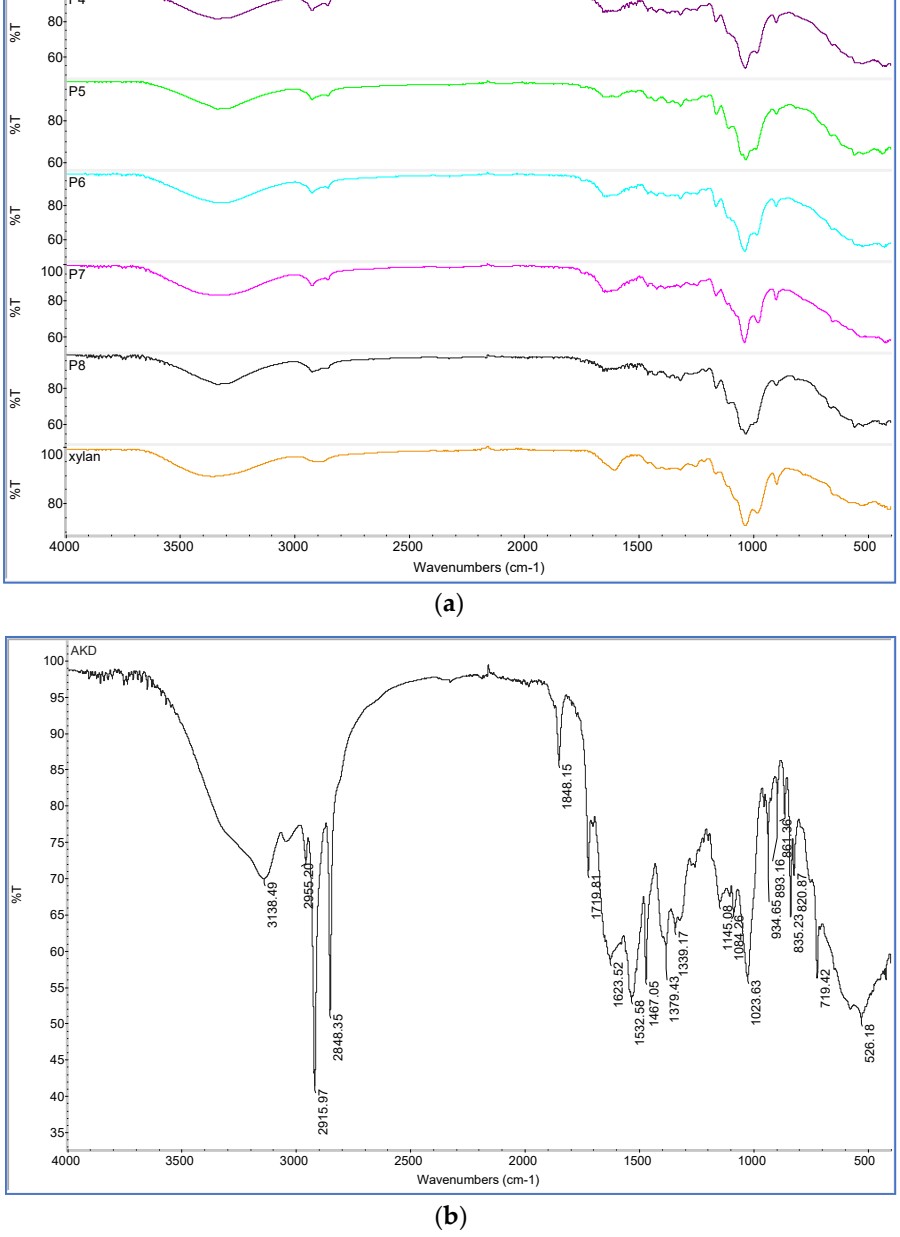

**Figure 5.** FT-IR spectra of: (**a**) xylan-coated paper (P4); xylan/0.2% AKD-coated paper (P5); xylan/0.5% AKD-coated paper (P6); xylan/1% AKD-coated paper (P7); xylan/1.5% AKD-coated paper (P8); (**b**) alkyl ketene dimer (AKD).

The sharp absorption peaks at 1848 and 1719 cm$^{-1}$ presented in AKD spectra (Figure 5b) are associated with stretching vibration of C=O and C=C groups from the lactone ring [57,58]. It can be observed that in the spectra from Figure 5a there is no AKD specific lactone ring peak at 1848 cm$^{-1}$. This proved that between xylan hemicellulose and AKD a covalent bond was formed, and AKD is not only physically adsorbed [54,59,60].

### 3.3. Water, Oil and Water Vapours Barrier Properties

Overall, as it can be observed from the results presented in Figure 6, the coated paper samples exhibit improved water barrier properties with increasing AKD content in coating formulation.

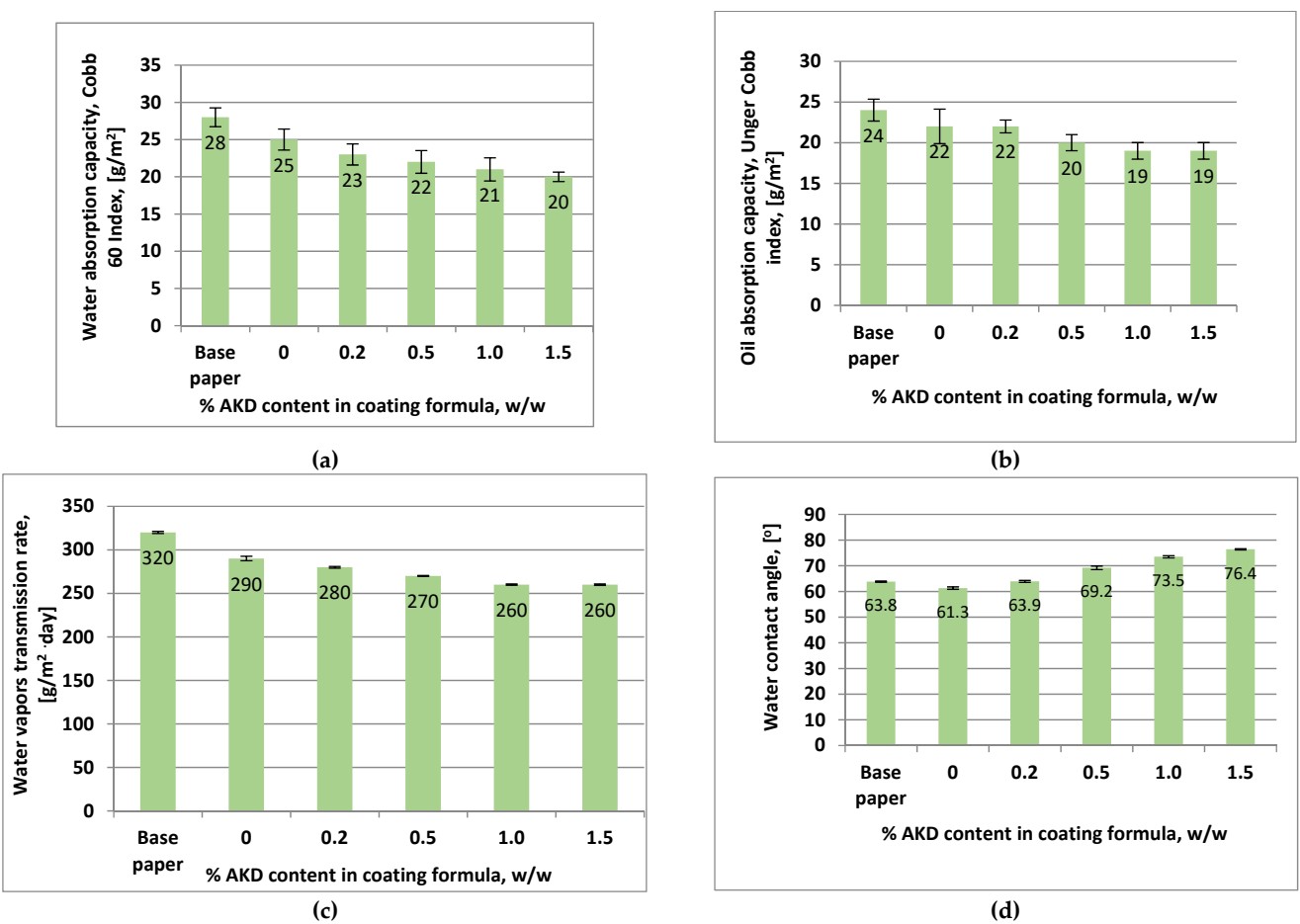

**Figure 6.** Barrier properties of coated papers samples with xylan and different content of AKDs in a single layer: (**a**) water absorption capacity; (**b**) oil absorption capacity; (**c**) water vapours transmission rate; (**d**) water contact angle. Error bars, often smaller than the plotted columns, indicate standard deviations.

On the one hand, the improving of barrier properties is a result of the hydrophobic nature of the AKD material as well as its ability for film forming. On the other hand, by heating of coated samples at high temperature the covalent bonds of AKD with xylan hemicelluloses are formed which impart hydrophobicity [29].

The water absorption capacity of coated samples with xylan is 11% lower than base paper compared with that of samples coated with xylan/1.5% AKD which was about 29% lower (Figure 6a) as result of high hydrophobicity of AKD.

Even if the xylan is highly hydrophilic its good ability for film forming leads to obtaining a compact structure with closed pores on the paper surface which blocks the air passing through the cellulose fibres network; therefore, it reduces the air permeability, being

45% lower compared with base paper (Figure 7). Furthermore, by adding of hydrophobic AKD the pore sealing is more effective and the air permeability is significantly improved for the paper samples coated with xylan/1.5% AKD being with 66% lower than that of base paper and 40% lower compared with that of xylan-coated samples [12].

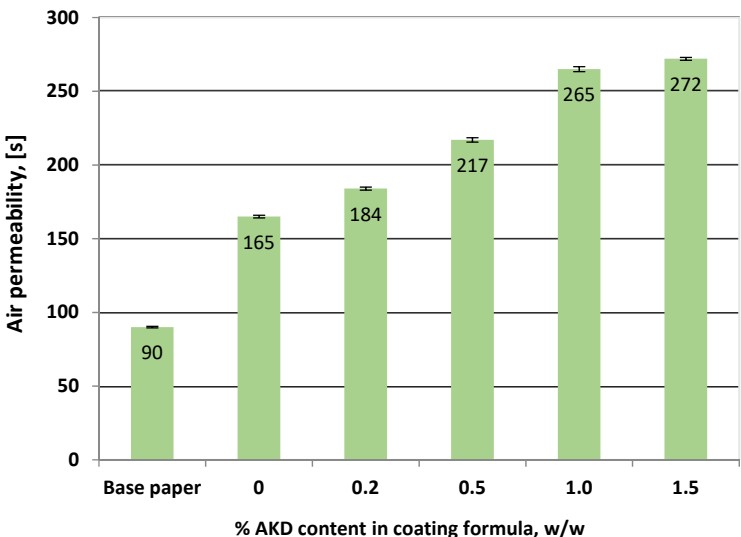

**Figure 7.** The air permeability of coated papers samples in a single layer with xylan and different contents of AKD. Error bars, often smaller than the plotted columns, indicate standard deviations.

The WVTR of xylan/1.5% AKD-coated papers was 19% lower than that of base paper, compared with only 11% improvement in the case of xylan-coated papers (Figure 6c). The slight increasing in WVTR is attributed to the presence of numerous micropores and cracks in the structure of coating layers at 1.5% AKD content, as it can be observed in SEM images from Figure 2e,f. The oil absorption of xylan/1.5% AKD-coated papers was improved compared with base paper and xylan-coated paper, being 21% and 14% lower, respectively (Figure 6b).

Favourable results were obtained for the water contact angle of xylan/1.5% AKD-coated paper which registered 76.4°, 20% higher than those of xylan-coated samples. However, it did not obtain a high hydrophobization level due to the fact that during the high temperature curing, AKD hydrolysis results in the conversion to ketone compounds. According to Marton's studies [61,62] the hydrolyzed products, which have already been converted to the ketone form, have no hydrophobization effect.

Having in view the results reported by Bildik et al. [42] which obtained contact angle of 85.36° and 92.16°, respectively, for paper treated with AKD only (about 0.5% and 1% AKD content to cellulose fibres) and the chemical structure of xylan, the obtained values for contact angle of coated papers with xylan/1.5% AKD in this study are considered appropriate for food paper package. A higher increase in contact angle and oil barrier level was obtained by Li et al. [54] for paper coated with emulsions based on xylan hemicelluloses and alkyl ketene dimer. In this case, for the coating amount of 6.11 g/m$^2$, the oil resistance and water contact angle of the paper were increased about 5 times and 66% compared with those of untreated samples [54].

Generally, the grammage of polymer coatings is an important parameter to develop and control the barrier properties of papers [63]. For the same coating amount, the water barrier properties of xylan/1.0% AKD-coated papers are only slightly improved as the number of coating layers is increased (Figure 8). As it can be observed, only WVTR and air permeability registered lower values (with 20% and 8% respectively) compared with the papers coated with a single layer (Figure 8b,c).

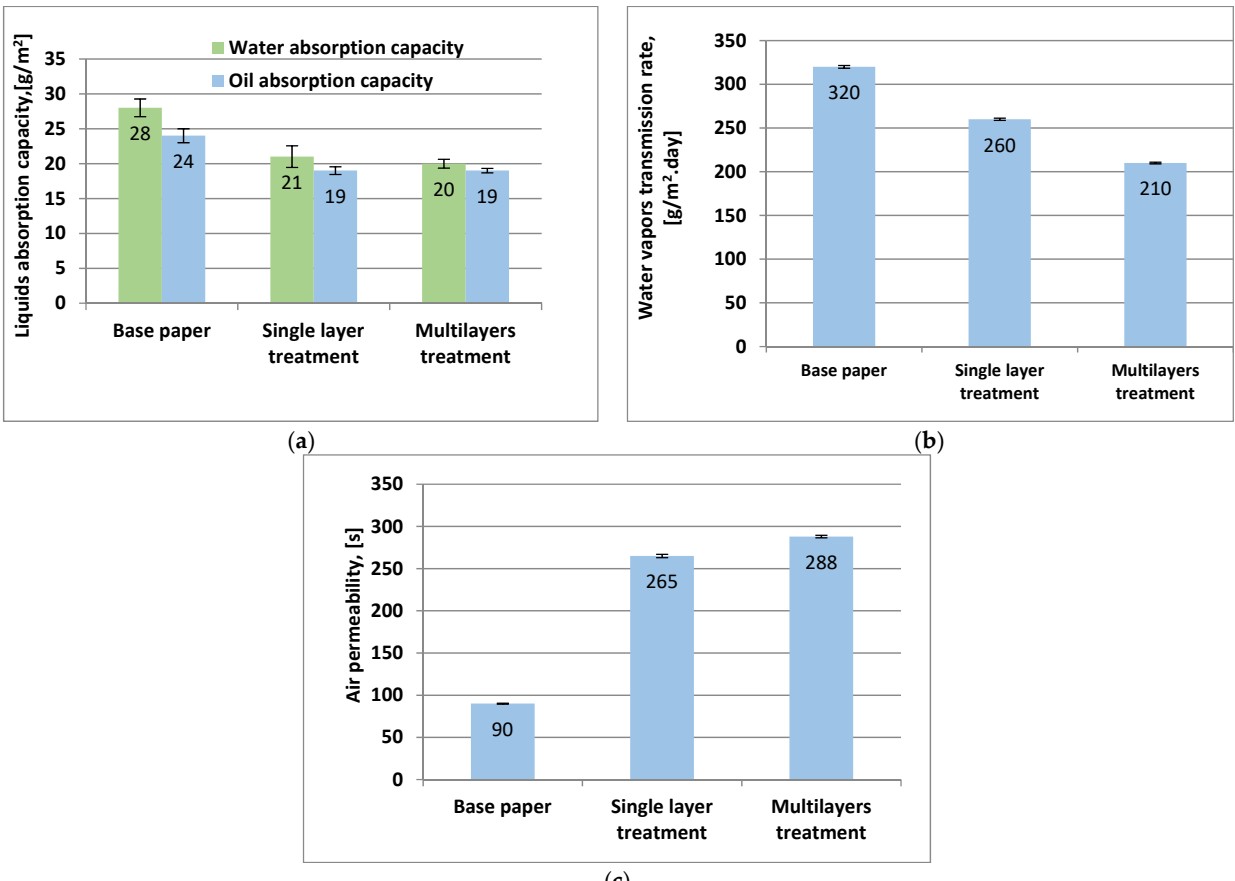

**Figure 8.** Barrier properties of coated paper samples with xylan/1% AKD in three layers: (**a**) water and oil absorption capacity; (**b**) water vapours transmission rate; (**c**) air permeability. Error bars, often smaller than the plotted columns, indicate standard deviations.

Through successive coatings the pores on the paper surface are closed, reducing the porosity and free superficial energy of the substrate [18,54,64]. This can lead to enhancing barrier properties, but due to of cracks and nonuniformities in the coating layer these properties are only less improved (see Figure 3b).

### 3.4. Mechanical Strength Properties

Bursting and tensile strength as well as the tearing resistance are considered important mechanical properties for packaging papers. During utilization, paper packages are subjected to repeated mechanical stretches, a certain level of strength is necessary. The addition of AKD led to the slight improvement of mechanical properties of the xylan-coated papers (Table 1). The dry/wet tensile and bursting strengths of xylan/1.5% AKD-coated papers were only 4 % and 6% higher compared with those of xylan-coated papers. The explanation could be that by grafting of AKD to the xylan chains the number of hydrogen bonds between xylan units is reduced. Consequently, the mechanical strength of paper coated with xylan and AKD decreased with increasing AKD content [53]. In addition, by applying on the base paper surface, the coatings penetrate the fibrous network to some extent. As a result, a weakening of interfibre bonds occurs which reduces the tensile strength of the paper sample [64].

**Table 1.** The mechanical strength properties of coated paper samples with xylan/AKD in a single layer.

| Properties | Base Paper | Coated Paper Samples | | | | |
|---|---|---|---|---|---|---|
| | | Xylan | Xylan + 0.2% AKD | Xylan + 0.5% AKD | Xylan + 1.0% AKD | Xylan + 1.5% AKD |
| Dry tensile index, N·m/g | 64 (*sdv.* 0.121) | 58 (*sdv.* 0.895) | 59 (*sdv.* 0.724) | 61 (*sdv.* 0.653) | 60 (*sdv.* 0.345) | 60 (*sdv.* 0.326) |
| Wet tensile index, kN·m/g | 9 (*sdv.* 0.545) | 9 (*sdv.* 0.795) | 9 (*sdv.* 0.698) | 9.3 (*sdv.* 0.594) | 9.5 (*sdv.* 0.659) | 9.6 (*sdv.* 0.428) |
| Bursting index, kPa·m²/g | 4.4 (*sdv.* 0.545) | 4.8 (*sdv.* 1.013) | 4.9 (*sdv.* 0.955) | 4.9 (*sdv.* 1.342) | 5.0 (*sdv.* 0.754) | 5.0 (*sdv.* 0.829) |

The results obtained show that three coating layers not significantly improves the mechanical strength of coated papers (Figure 9a,b).

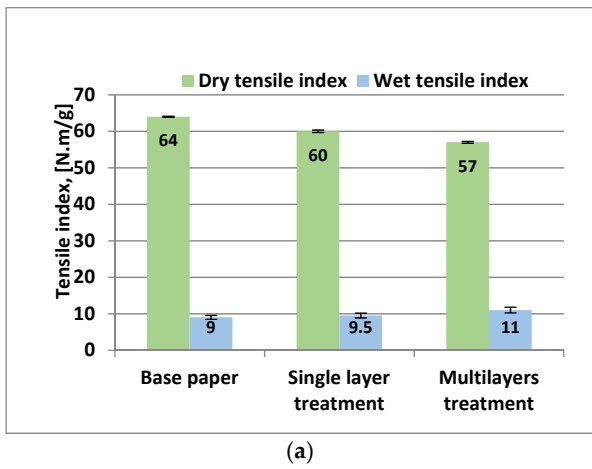

(**a**)

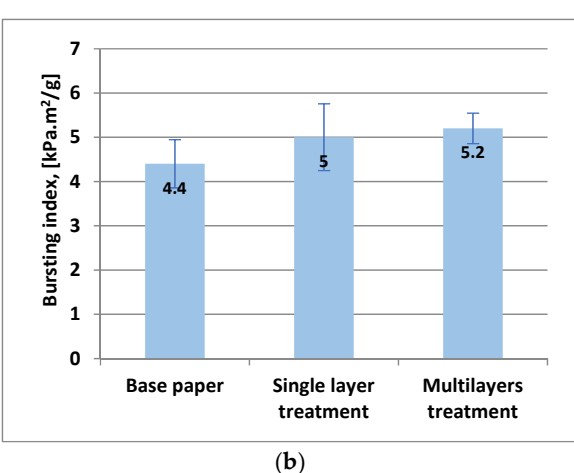

(**b**)

**Figure 9.** The mechanical strength properties of coated paper samples with xylan/1% AKD in three successive layers: (**a**) dry and wet tensile strength; (**b**) bursting strength. Error bars, often smaller than the plotted columns, indicate standard deviations.

*3.5. Sub*

The slight improvement or deterioration of mechanical strength of coated papers, both ways in single and three successive layers can be explained by the microstructure of coating layers. The appearance of defects and cracks in the coating layer structure led to deterioration of dry tensile strength of coated papers [41].

**4. Conclusions**

In this work, xylan hemicellulose, currently available in nature and underutilized, was chemically modified via esterification in a simple procedure using alkyl ketene dimer. This is a common paper sizing agent AKD, commercially available, with a low-cost and is biodegradable.

The dispersions of xylan/AKD were used to obtain coated paper with improved barrier properties. The structural analysis by SEM and FT-IR spectroscopy highlighted the presence of keto-esters functional groups by absorption peaks at 1602 cm$^{-1}$–1733 cm$^{-1}$. As result, the covalent bonds between xylan particles and AKD were formed that impart hydrophobicity with the effect of improving barrier properties of coated papers. Consequently, the water and oil absorption of paper samples coated in a single layer (4.5 g/m²) with xylan hemicelluloses/1% AKD were 25% and 32% lower than those of uncoated paper, respectively. For the coated paper with three successive layers, WVTR was enhanced compared with that of base paper, being 35% lower. In comparison with single layer coated

paper, WVTR was only slightly improved being about 20% lower. The duration of air passing through coated papers was over three times higher comparing with base paper and only 8% higher comparing with paper samples coated in a single layer. This is the result of cracks and unevenness present in the coating layer.

The mechanical strength properties are only less improved as content of AKD in coating dispersion is increased. This is the effect of AKD grafting on the xylan chains when the number of hydrogen bonds between xylan units which are responsible by strength improving, is reduced. In addition, as an effect of coating treatment, the polymer dispersion penetrates into the interfibre network to some extent, diminishing their break tensile performance.

These findings are promising and indicate that xylan hemicelluloses/AKD coatings can be an appropriate alternative for synthetic polymers in food packaging and extend the applications area of xylan hemicellulose. Even if there are few results reported on hemicellulose modification with AKD, this procedure is simple and suitable to apply due the large availability and high stability of AKD compared with other chemical modification methods that use acetic anhydride or enzyme. Moreover, the packaging papers coated with xylan/AKD can be easily recycled in a papermaking chain.

**Author Contributions:** Conceptualization, P.N. and M.R.; methodology, P.N. and M.R.; software, P.N.; validation, P.N., M.R., A.C.C. and A.V.D.B.; investigation P.N., M.R., A.C.C. and A.V.D.B.; writing—original draft preparation, P.N.; writing—review and editing, P.N. and M.R.; visualization, P.N., M.R., A.V.D.B. and A.C.C.; supervision, P.N. All authors have read and agreed to the published version of the manuscript.

**Funding:** This research was funded by the Ministry of Research, Innovation, and Digitization, CNCS—UEFISCDI, Romania grant number PN-III-P4-PCE-2021-0714.

**Institutional Review Board Statement:** Not applicable.

**Informed Consent Statement:** Not applicable.

**Data Availability Statement:** Data are provided in the paper.

**Acknowledgments:** This work was supported by a grant of the Ministry of Research, Innovation and Digitization, CNCS—UEFISCDI, project number PN-III-P4-PCE-2021-0714, within PNCDI III.

**Conflicts of Interest:** The authors declare no conflict of interest.

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
