# Peer review of "Improving Barrier Properties of Xylan-Coated Food Packaging Papers with Alkyl Ketene Dimer"

_sustainability, doi:10.3390/su142316255_

Round 1
Reviewer 1 Report
The manuscript can be accepted with a minor revision. Please consider the following comments:
1. Check all the citations, especially in the introduction; several citations are placed between two dots.
2. check the format of the journal
3. In Figure 5, decrease the thickness of the spectra
4. there are many grammatical errors, especially in the introduction. Please check all.
Author Response
Dear reviewer,
Thank you very much for your useful observations, comments and recommendations. We deeply appreciate your evaluation.Your fruitful comments helped us to improve the overall quality of the manuscipt.
Please, find bellow, point by point, the response at your comments:
Reviewer comments
- Check all the citations, especially in the introduction; several citations are placed between two dots.
- check the format of the journal
- In Figure 5, decrease the thickness of the spectra
- there are many grammatical errors, especially in the introduction. Please check all.
Response
Whole manuscript was depply checked and revised. All citations were revised and 14 new references were introduced. Figure 5 was changed with new one with better resolution.
All the modifications are highlighted in the revised manuscript with red color.

Reviewer 2 Report
Dear Authors: Thank you for your manuscript discussing the mechanical and barrier properties of paper substrates coated with xylan and AKD.
The initial idea sounds interesting, but the story as described is not appealing enough. The introduction does not refer to any previous works in the field or refer to potential of commercialization or enhanced sustainability aspects of the proposed solution.
The experimental plan is rather simple to understand although I would include a reference in the discussion: paper substrates coated with AKD only. The resolution of the figures & graphics is very low, and the graphics require enhanced scientific quality. Too many significant digits, high standard deviations....no clear change in the properties and no scientific discussion as to why this is happening, and how to address the observed pitfalls.
With contact angle values of ca. 76-77 degree, you do not really have strong hydrophobicity to claim. The increase or decrease of properties observed do not look significant enough to be discussed. Please run statistical analysis.
The work, as written, does not highlight its intellectual merits: how and why is this study new? in what ways are the results significant and important?
The manuscript also needs strict English revisions and please, check the typos in the cited references.
Author Response
Dear reviewer,
Thank you very much for your useful observations, comments and recommendations. We deeply appreciate your evaluation. Your fruitful comments helped us to improve the overall quality of the manuscript. Please, find bellow, point by point, the response at your comments:
Reviewer comments
The initial idea sounds interesting, but the story as described is not appealing enough. The introduction does not refer to any previous works in the field or refer to potential of commercialization or enhanced sustainability aspects of the proposed solution.
Response
Whole manuscript was deeply checked and revised. The introduction section was revised with references at previous work in the field and the sustainability aspects of proposed solution. 14 new references were introduced.
All the modifications are highlighted in the revised manuscript with red color.
Reviewer comments
The experimental plan is rather simple to understand although I would include a reference in the discussion: paper substrates coated with AKD only.
Response
A paper substrate coated with AKD only was not considered, due the fact that it is know the hydrophobisation effect of AKD only on the cellulose fibres. To emphasize this aspect, within the revised manuscript, were introduced for comparison the results on contact angle obtained by Bildik et al. [42] for the paper treated with AKD at the same content which was used in xylan dispersions. (about 0.5% and 1% AKD content to cellulose fibres and contact angle was 85.36° and 92.16°, respectively)
Reviewer comments
The resolution of the figures & graphics is very low, and the graphics require enhanced scientific quality. Too many significant digits, high standard deviations....no clear change in the properties and no scientific discussion as to why this is happening, and how to address the observed pitfalls.
Response
All the figures&graphics were reconsidered in the revised manuscript as well as all the discussions on the results interpretation. The Methods section was completed with the number of tests for each analysis and standard deviation was clearly highlighted on the graphs and tables.
Reviewer comments
With contact angle values of ca. 76-77 degree, you do not really have strong hydrophobicity to claim. The increase or decrease of properties observed does not look significant enough to be discussed. Please run statistical analysis.
Response
It is not obtained a high hydrophobization level due to the fact that during the temperature curing (110°C), AKD hydrolysis results in the conversion to ketone compounds. The hydrolyzed products, which have been converted to the ketone form, have no hydrophobization effect. The FT-IR analyse confirmed the keto-esters formation. As result, the low hydrophobisation level was obtained. In addition, the coating dispersions were water based.
In this case a number of 10 measurements were developed standard deviation was clearly highlighted on the graph. Obviously, in the future work, we intend to run statistical analysis, also.
Reviewer comments
The work, as written, does not highlight its intellectual merits: how and why is this study new? in what ways are the results significant and important?
The manuscript also needs strict English revisions and please, check the typos in the cited references.
Response
Whole the manuscript was revised and some sections were re-considered to highlight the practical and scientifical importance of described topic.
In the literature there are many studies which report the utilisation of xylan as film in food packaging. The research on utilisation of xylan hemicelluloses in paper coating are still infancy. In addition, the AKD is widely used in paper sizing, but its utilisation to hydrophobisation of hemicellulose is not reported. The commercial availability of AKD strengthens its use in the process of xylan functionalization by chemical modification and thus, a new type of hydrophobic xylan could commercially produce.
The manuscript was also revised regarding the English.

Reviewer 3 Report
The manuscript entitled "Improving barrier properties of xylan coated food packaging papers with alkyl ketene dimer" examined the effect of xylan coating and alkyl ketene dimer on structural, barrier and mechanical properties of paper. Generally, the manuscript is well written, however some improvements are needed. My suggestions are listed below:
1. Clearly state the main goals of your study at the end of the Introduction part.
2. Which statistical tests were performed to prove the significance of the results? This should be described in methods.
3. Sections 3.3 and 3.4: The results are presented only on figures/table. Support the claim "....are improved" by numerical values or add the percentage of improvement.
Author Response
Dear reviewer,
Thank you very much for your useful observations, comments and recommendations. We deeply appreciate your evaluation. Your fruitful comments helped us to improve the overall quality of the manuscript. Please, find bellow, point by point, the response at your comments:
Reviewer comments
- Clearly state the main goals of your study at the end of the Introduction part.
Response
Whole manuscript was deeply checked and revised. The introduction section was revised with references at previous work in the field and the sustainability aspects of proposed solution. The main goals of the work were clearly described. 14 new references were introduced.
All the modifications are highlighted in the revised manuscript with red color.
Reviewer comments
- Which statistical tests were performed to prove the significance of the results? This should be described in methods.
Response
The Methods section was completed with the number of tests for each analysis. The average values were disscused and standard deviation was clearly highlighted on the graphs and tables.
Reviewer comments
- Sections 3.3 and 3.4: The results are presented only on figures/table. Support the claim "....are improved" by numerical values or add the percentage of improvement.
Response
These sections were re-considered in the revised manuscript and numerical values were added to support the plotted and presented results.

Reviewer 4 Report
Nechita et al, have worked on improving the barrier properties of paper via the coating of xylan together with alkyl ketene dimer. In their work, they reported successful barrier property improvements. The work is attractive and can be published in the journal sustainability provided the authors do significant improvements in revised version. My comments on the manuscript are mentioned below:
1. The authors should enrich the introduction section with a healthy discussion on processing steps along with its advantages and possible disadvantages.
2. What is the novelty of this work? Please clearly define it at the end of the introduction section.
3. The statement given in lines from 42 to 44 requires citations.
4. Please explain the scientific reason as to why barrier properties increased with the addition of AKD? As the SEM image in Figure 3(b) shows, there are still some porous spaces. Please justify the barrier improvement in this context.
5. Some of the latest literature can be added in the intro part for better discussion.
Such as:
https://doi.org/10.3390/ma15031046,
https://doi.org/10.3390/membranes12070701,
https://doi.org/10.3390/polym13040568
6. Section 3 should be Results and Discussion rather than just Results
Author Response
Dear reviewer,
Thank you very much for your useful observations, comments and recommendations. We deeply appreciate your evaluation.Your fruitful comments helped us to improve the overall quality of the manuscript. Please, find bellow, point by point, the response at your comments:
Reviewer comments
The authors should enrich the introduction section with a healthy discussion on processing steps along with its advantages and possible disadvantages.
Response
Whole manuscript was deeply checked and revised. The introduction section was revised with references at the sustainability aspects as well as on the advantages and disadvantages of proposed solution. The main goals of the work were clearly described. 14 new references were introduced.
All the modifications are highlighted in the revised manuscript with red color.
Reviewer comments
What is the novelty of this work? Please clearly define it at the end of the introduction section.
Response
In the literature there are many studies which report the utilisation of xylan as film in food packaging. The research on utilisation of xylan hemicelluloses in paper coating are still infancy. In addition, the AKD is widely used in paper sizing, but its utilisation to hydrophobisation of hemicellulose is not reported. The commercial availability of AKD strengthens its use in the process of xylan functionalization by chemical modification and thus, a new type of hydrophobic xylan could commercially produce.
These aspects were defined in the Introduction Section of revised manuscript.
Reviewer comments
The statement given in lines from 42 to 44 requires citations.
Response
Whole manuscript was deeply checked and revised, and 14 new references were introduced.
Reviewer comments
Please explain the scientific reason as to why barrier properties increased with the addition of AKD? As the SEM image in Figure 3(b) shows, there are still some porous spaces. Please justify the barrier improvement in this context.
Response
In the revised manuscript the scientifical explanation regarding the improving barrier properties with addition of AKD was introduced in the Introduction section and in the Results and discussion section. The interpretation of obtained results was reconsidered in the revised manuscript. All the modifications are highlighted in the revised manuscript with red color.
Reviewer comments
Some of the latest literature can be added in the intro part for better discussion.
Response
In the revised manuscript two of the indicated references (doi:10.3390/polym11030560 , https://doi.org/10.3390/ma15031046) were introduced in the section Introduction, and are highlighted with green color (ref, no. 6 and no.35).
Reviewer comments
Section 3 should be Results and Discussion rather than just Results
Response
In the revised manuscript this modification was included.

Round 2
Reviewer 2 Report
Significant improvements have been made to the manuscript, but the structure and language of the text must be improved.
The air permeability was increased after coating, yet the authors concluded in a general way that the barrier properties of their substrates were improved. The conclusion is inconsistent with the results.
Changes in water and oil absorption, and WVTR are also not significant enough to be highlighted. Same for the mechanical properties.
The significance of the results should be higher for publication.
Author Response
Dear Reviewer,
Thank you very much for your useful observations, comments and recommendations that helped us to improve the overall quality of the manuscript. Please, find bellow, point by point, the response at your comments:
Comment: The air permeability was increased after coating, yet the authors concluded in a general way that the barrier properties of their substrates were improved. The conclusion is inconsistent with the results.
- in the manuscript is the highlighted this improving as follow…..” Even if the xylan is high hydrophilic its good ability for film forming leads to obtaining a compact structure with closed pores on the paper surface which blocks the air passing through the cellulose fibers network; therefore, it reduces the air permeability, being 45% lower comparing with base paper (Figure 7). Furthermore, by adding of hydrophobic AKD the pore sealing is more effective and the air permeability is significantly improved for the paper samples coated with xylan/1.5%AKD being with 66% lower than that of base paper and 40% lower comparing with that of xylan coated samples”….
Comment: Changes in water and oil absorption, and WVTR are also not significant enough to be highlighted. Same for the mechanical properties.
- Indeed, the improving of liquids absorption and WVTR are not significantly improved as well as the mechanical strength of coated paper. This is emphasized in the manuscript as follow:….. “The slightly increasing of WVTR is attributed to the presence of numerous micropores and cracks in the structure of coating layers at 1.5%AKD content, as it can be observed in SEM images from Figures 2 (e) and (f)”…… and …” However it is not obtained a high hydrophobization level, due to the fact that during the high temperature curing, AKD hydrolysis results in the conversion to ketone compounds. According to Marton’s studies [55, 56] the hydrolyzed products, which have already been converted to the ketone form, have no hydrophobization effect…..” and ….” The dry/wet tensile and bursting strengths of xylan/1.5% AKD coated papers were only 4 % and 6% higher comparing with those of xylan coated papers. The explanation could be that, by grafting of AKD to the xylan chains the number of hydrogen bonds between xylan units is reduced. Consequently, the mechanical strength of paper coated with xylan and AKD decreased with increasing of AKD content [53]. In addition, by applying on base paper surface, the coatings penetrate the fibrous network to some extent. As results, a weakening of interfiber bonds occur which reduces the tensile strength of paper sample [58].”….
The conclusions section was re-considered according to your recommendations and observations.
The manuscript was also checked carefully regarding the English language, by a specialised person.
The authors

Reviewer 4 Report
The authors have significantly improved the overall quality of the manuscript, I think, the manuscript is acceptable for the publication in its present form
Author Response
Dear reviewer,
The authors have significantly improved the overall quality of the manuscript, I think, the manuscript is acceptable for the publication in its present form
R. Thank you very much for your useful observations, comments and recommendations. We deeply appreciate your evaluation.Your fruitful comments helped us to improve the overall quality of the manuscript.
The authors